# ANATOMY-DT: A CROSS-DIFFUSION DIGITAL TWIN FOR ANATOMICAL EVOLUTION

## ABSTRACT

Accurately modeling the spatiotemporal evolution of tumor morphology from baseline imaging is a pre-requisite for developing digital twin frameworks that can simulate disease progression and treatment response. Most existing approaches primarily characterize tumor growth while neglecting the concomitant alterations in adjacent anatomical structures. In reality, tumor evolution is highly non-linear and heterogeneous, shaped not only by therapeutic interventions but also by its spatial context and interaction with neighboring tissues. Therefore, it is critical to model tumor progression in conjunction with surrounding anatomy to obtain a comprehensive and clinically relevant understanding of disease dynamics. We introduce a mathematically grounded framework that unites mechanistic partial differential equations (PDEs) with differentiable deep learning. Anatomy is represented as a multi-class probability field on the simplex and evolved by a cross-diffusion reaction–diffusion system that enforces inter-class competition and exclusivity. A differentiable implicit–explicit (IMEX) scheme treats stiff diffusion implicitly while handling nonlinear reaction and event terms explicitly, followed by projection back to the simplex. To further enhance global plausibility, we introduce a topology regularizer that simultaneously enforces centerline preservation and penalizes region overlaps. The approach is validated on synthetic datasets (Voronoi, Vessel) and a clinical dataset (UCSF-ALPTDG brain glioma). On synthetic benchmarks, our method achieves state-of-the-art accuracy (e.g., Voronoi-DSC: $95.70 \pm 0.30$ and Vessel-DSC: $71.14 \pm 0.25$) while preserving topology, and also demonstrates superior performance on the clinical dataset (UCSF-DSC: $65.37 \pm 0.35$). By integrating PDE dynamics, topology-aware regularization, and differentiable solvers, this work establishes a principled path toward anatomy-to-anatomy generation for digital twins that are visually realistic, anatomically exclusive, and topologically consistent. Code will be made available upon acceptance.

## 1 INTRODUCTION

Modeling the temporal evolution of anatomical structures is a central challenge in computational oncology and medical image analysis (Ren et al., 2022; Lachinov et al., 2023). Clinical imaging protocols routinely capture patient scans before and after treatment, or during disease progression, yet predictive models that generate plausible future anatomy from a baseline scan remain limited. The main reasons being scarce training datasets and lack in the existing mechanisms to enforce temporal consistency and anatomical plausibility in biological growth dynamics. There is a need to develop computational surrogates for individual patients that simulate trajectories of disease growth and therapeutic response, thereby informing personalized treatment planning and clinical decision support (Katsoulakis et al., 2024; Kuang et al., 2024).

A potential solution to predicting growth trajectories is through generative AI approaches. Most existing generative modeling techniques in medical imaging, such as conditional generative adversarial networks (GANs) and diffusion models (Ho et al., 2020; Dhariwal & Nichol, 2021; Armanious et al., 2020), operate primarily generate images from gaussian noise without considering any treatment paradigms as input. While these methods produce visually realistic outputs, they often fail to guarantee structural plausibility. To ensure structure correctness, several methods have been proposed that use conditioning mechanisms to generate images (Zhang et al., 2023; Zhao et al., 2023). Using anatomies as control has been shown to improve medical image generation (Bhat-

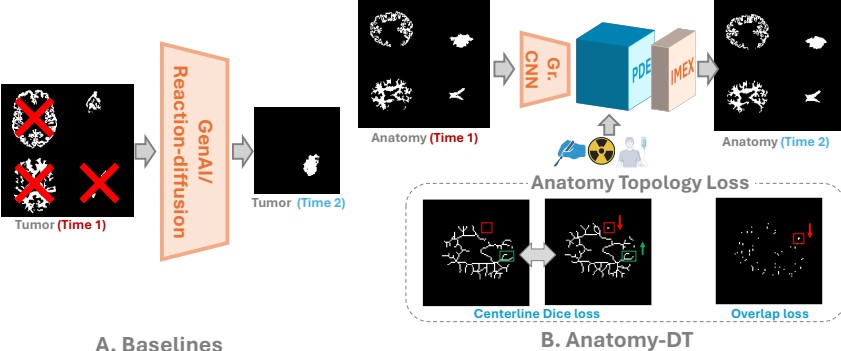

Figure 1: Existing baselines model only tumor growth and do not consider adjacent anatomy changes. Our proposed method, Anatomy-DT, models both anatomy and tumor growth conditioned on different treatment paradigms.

tacharya et al., 2024; 2025a). Recent methods can generate anatomically accurate post-treatment images from pre-treatment scans (Bhattacharya et al., 2025b; Liu et al., 2025) when conditioned with patient demographics, genomic markers, etc. The major limitations of these methods are that they are static in nature and cannot handle heterogenous inputs i.e., pre-treatment, pre-operative, post-operative scans, varying treatment schemes, genomic mutations. In contrast, mechanistic models of tumor growth and tissue dynamics, often formulated as reaction–diffusion partial differential equations (PDEs) (Martens et al., 2022; Yin et al., 2019; Metzcar et al., 2024), provide interpretable dynamics grounded in biology. These models capture important features such as infiltration along tissue interfaces or mass effect on surrounding tissue. However, they are typically restricted to single-class tumor representations, lack explicit coupling with surrounding anatomy, and cannot be easily coupled with deep learning frameworks for end-to-end training. Moreover, they rarely account for global topological invariants, such as the connectivity of ventricles or the tree structure of vessels, which are critical to ensure clinically valid anatomical trajectories. Recently proposed works ((Bhattacharya et al., 2025a) and Gupta et al. (2024)) take topological constraints into account to ensure topological consistency in the generated images(Bhattacharya et al., 2025a; Gupta et al., 2024). With the emergence of digital twins, mechanistic models conditioned on diverse treatment modalities have re-emerged as an important paradigm. Clinically, however, these treatments exert influence not only on the tumor itself but also on adjacent anatomical structures. Capturing the coupled evolution of multiple anatomies is particularly challenging, as it requires maintaining both spatial coherence and topological consistency across interacting tissues. Topology-constrained mechanistic models provide a principled framework to address this challenge, ensuring preservation of anatomical topology while enabling clinically faithful simulations.

In this work, we introduce a novel framework, Anatomy-DT, an anatomy digital twin that bridges the gap between mechanistic PDEs and data-driven generative models. Our approach represents anatomy as a multi-class probability field $u(x,t) \in \Delta^{K-1}$, where $K$ denotes the number of tissue classes and $\Delta^{K-1}$ is the probability simplex. The temporal evolution of this field is governed by a *cross-diffusion reaction–diffusion system*, which naturally enforces competition between classes and exclusivity of tissue regions. Concretely, the governing PDE is formulated as

$$\partial_t u_i(x,t) \;=\; \nabla \cdot \left( \sum_{j=1}^{K} D_{ij}(u)\, \nabla u_j(x,t) \right) \;+\; R_i(u)$$

where $D_{ij}(u)$ encodes both self-diffusion ($i = j$) and cross-diffusion ($i \neq j$) coefficients, and $R_i(u)$ represents local reaction or growth terms (e.g., proliferation, atrophy, or event-driven changes). Such systems generalize classical reaction–diffusion equations Murray (2001); Turing (1990); Yankeelov et al. (2015); Jarrett et al. (2018) and have been studied extensively in multi-species dynamics Shigesada et al. (1979); Chen et al. (2021); Burger et al. (2020). In the anatomical setting, this structure enforces sharp, mutually exclusive tissue boundaries without resorting to ad-hoc normalization.

To achieve stable numerical integration within deep learning pipelines, we develop a differentiable implicit–explicit (IMEX) scheme Ascher et al. (1995); Hundsdorfer & Verwer (2013) that treats the stiff diffusion operator implicitly, while handling the nonlinear reaction and event terms explicitly. After each update, the solution is projected back onto the simplex, ensuring tissue probability conservation and anatomical plausibility. This design yields a numerically stable PDE layer that is fully compatible with backpropagation and scalable to medical images.

To further guarantee anatomical plausibility beyond local PDE dynamics, we incorporate global topological priors. Anatomical structures are inherently complex, and their organization becomes further disrupted during tumor growth owing to processes such as angiogenesis, stromal remodeling, extracellular matrix degradation, edema formation, and mass effect, which collectively alter both local microarchitecture and global tissue geometry. In this work, we focus on modeling how anatomical structures deform in response to tumor growth. We propose a formulation that enforces topological correctness by combining centerline consistency for complex anatomical structures with a no-overlap constraint across different anatomies. Topological regularization has been successfully applied to image analysis and generative models Clough et al. (2020); Hofer et al. (2019), but to our knowledge, this is the first integration with a cross-diffusion PDE generative layer. Taken together, the combination of cross-diffusion PDE dynamics, differentiable IMEX integration, and topology-aware regularization constitutes a principled mathematical formulation for generative anatomy-to-anatomy modeling.

We experiment on two synthetic datasets, demonstrating Anatomy-DT's ability to reproduce both local tissue proliferation and global topology preservation. Beyond synthetic validation, the framework can be directly applied to clinical datasets such as brain MRIs, where baseline masks are evolved into follow-up masks conditioned on treatment variables. By unifying mechanistic priors with differentiable generative modeling, our method lays the foundation for digital twins that are not only visually realistic, but also anatomically and topologically consistent.

Our contributions are as follows: a) We propose Anatomy-DT, a cross-diffusion reaction–diffusion PDE on the simplex for anatomy-to-anatomy generation, introducing inter-class proliferation as a novel generative mechanism, b) We design a differentiable IMEX solver that implicitly handles stiff diffusion, explicitly treats nonlinear terms, and projects onto the simplex to ensure stability, probability conservation, and anatomical plausibility within end-to-end training, c)We introduce topology-preserving regularization by enforcing centerline consistency and inter-anatomy no-overlap, providing structural guarantees absent in standard generative models and d) We demonstrate the effectiveness of the proposed method on a clinical dataset for post-treatment tumor and multiple-anatomy evolution prediction.

Together, these contributions lead to a tumor and multi-anatomy growth modeling paradigm constrained on different treatment types.To the best of our knowledge, this is the first approach that integrates PDE dynamics and topological constraints into digital twin frameworks.

## 2 METHODS

### 2.1 BACKGROUND

Diffusion-based PDEs are widely used to model spatio-temporal processes in biology particularly in tumor growth, spread of signaling molecules, etc. In biological application, these diffusion PDEs are used in conjunction with reaction representation such as proliferation, chemotherapy kill, etc. (**?**). These systems are termed as *Reaction Diffusion PDEs*. However, such methods typically focus on modeling the growth of a single class, such as a tumor, while neglecting the concurrent structural and functional changes in surrounding anatomies. Here, we provide a brief background of different diffusion methods for multi-class setting ($k$).

**Self-diffusion.** For class $k$, the term $\nabla \cdot (D_k(x)\nabla p_k)$ models independent spreading, such as edema expansion along white-matter tracts, where $D_k(x)$ denotes a spatially varying diffusivity and $p_k$ the probability or density of class $k$. This method smooths local concentrations of different classes but does not enforce interactions between them (Swanson et al., 2000).

**Cross-diffusion.** The term $-\nabla \cdot (\chi_{kj}(x)\, p_k \nabla p_j)$ encodes inter-class interactions in which the spatial gradients of class $i$ influence the diffusion of class $j$, where $i, j \subset k$. This method enforces anatomical exclusivity by avoiding inter-class diffusion (Vanag & Epstein, 2009).

Figure 2: **Anatomy-DT architecture.** Our proposed method has three primary components: a) A Growth CNN that learns the residual patterns of anatomy growth, b) the cross-diffusion PDE which models multi-anatomy evolution and c) a topology loss function preserving the anatomical structures.

**Treatment terms.** In tumor growth modeling, in addition to the diffusion terms, there are also reaction terms like proliferation represented as $r_k$ and discrete interventions such as surgery or radiotherapy represented as $s_k$. The reaction diffusion model can be represented as

$$\frac{\partial p_k}{\partial t} = \nabla \cdot \big( D_k(x) \nabla p_k \big) + r_k(p_k) + s_k. \tag{1}$$

where $p_k$ is the probability of each class.

## 2.2 STATE REPRESENTATION

In this work, we propose a system that models tumor and anatomy growth. We first define how anatomical structures ($\mathbb{A}_k$) are represented over time $t \in [0, T]$. Instead of using discrete segmentation masks, which are non-differentiable and cannot capture uncertainty, we adopt a probabilistic formulation where each pixel/voxel $x$ is described as a distribution over anatomical classes. This formulation is differentiable and captures both relatively stable anatomical structures and evolving pathologies like tumors.

**Definition 1** (Anatomical State). *We define the anatomical state at time $t$ as a multi-class probability field*

$$p : \Omega \times [0, T] \to \Delta^{K-1}, \qquad p(x, t) = (p_1(x, t), \dots, p_K(x, t)),$$

*where $\Omega \subset \mathbb{R}^D$ denotes the image domain and $\Delta^{K-1}$ is the $(K-1)$-simplex.*

*Remark* 1. The simplex constraint,

$$p_k(x, t) \geq 0, \quad \sum_{k=1}^{K} p_k(x, t) = 1,$$

ensures mutual exclusivity of tissue classes. This means that even though the model operates in a relaxed probability space, every pixel is assigned to exactly one dominant class at inference. Such a design allows us to reason jointly about multiple tissues while maintaining anatomical plausibility.

## 2.3 ANATOMY DIGITAL TWIN

Anatomy-DT models anatomy and tumor growth across different timepoints given that the patient is subjected to certain treatments. Having established the representation in the previous sub-section, we now specify the model governing temporal evolution of the different anatomies and tumors. We build on cross-diffusion reaction–diffusion systems, which are widely used in biology and ecology to capture competition between different interacting categories. In this particular context, these models are useful as they not only capture the independent proliferation of each class, but also captures spatial competition and exclusivity between different classes. To enhance expressive power, we further

integrate a Growth CNN, which learns residual corrections to the PDE dynamics and better aligns simulated trajectories with observed imaging data.

In conjunction with cross-diffusion and proliferation terms, we propose to incorporate clinical interventions, such as surgical resections, radiotherapy, or chemotherapy (in any combination). To this end, we augment the cross–diffusion reaction–diffusion system with the per-class treatment term $s_k$. These terms allow the framework to capture both the natural dynamics of tissue competition and the discontinuities introduced by medical interventions, resulting in a more faithful representation of patient-specific tumor–anatomy evolution.

**Definition 2** (Cross-Diffusion PDE model). *The dynamics of $p(x, t)$ are governed by a multi-class cross-diffusion reaction–diffusion system. For each class $k$, the PDE is represented as:*

$$\partial_t p_k(x, t) = \nabla \cdot \left( D_k(x) \nabla p_k(x, t) - \sum_{j \neq k} \chi_{kj}(x) \, p_k(x, t) \nabla p_j(x, t) \right) + r_k(p; C, U, x, t) + s_k(x, t). \tag{2}$$

*Remark* 2. Here, $D_k(x)$ encodes tissue-specific anisotropic diffusion, $\chi_{kj}(x)$ regulates how classes compete for space, $r_k$ models growth or atrophy, and $s_k$ captures interventions such as surgery or radiation. This general formulation integrates continuous biophysical dynamics with discrete medical events, making it well suited for constructing patient-specific digital twins.

For the tumor class, $r_{\text{tumor}} = \alpha(x) \, p_{\text{tumor}} \left( 1 - \frac{p_{\text{tumor}}}{\kappa(x)} \right)$, where $\alpha(x)$ is the local growth rate and $\kappa(x)$ is the carrying capacity. This classical logistic growth term captures exponential proliferation at low density and saturation at high density, reflecting biological growth limits.

## 2.4 NUMERICAL INTEGRATION VIA IMEX SCHEME

Directly integrating equation 2 is numerically unstable due to stiffness in the diffusion terms. To address this, we design a differentiable solver based on an implicit–explicit (IMEX) scheme. This splitting stabilizes integration while preserving sharp intervention effects such as resections. We first decompose the PDE into stiff and non-stiff parts $\partial_t p = F_{\text{stiff}}(p) + F_{\text{nonstiff}}(p)$, with diffusion and cross-diffusion in $F_{\text{stiff}}$, and reaction and intervention terms in $F_{\text{nonstiff}}$. Then we perform IMEX Time Discretization. The scheme is given by $\frac{p^{n+1} - p^n}{\Delta t} = F_{\text{stiff}}(p^{n+1}) + F_{\text{nonstiff}}(p^n)$. Implicit updates require solving $(I - \Delta t D_k \nabla^2) p_k^{n+1} = \text{rhs}$, which we approximate with $m$ Jacobi iterations. Unrolling these iterations yields a solver compatible with backpropagation, enabling gradients to flow through temporal dynamics. We observe that the IMEX scheme provides unconditional stability with respect to step size for the diffusion terms, while explicit updates preserve the discontinuities induced by clinical interventions. Projection back to the simplex ensures feasibility of the anatomical state throughout training.

## 2.5 TOPOLOGY-PRESERVING REGULARIZATION

While PDE dynamics ensure plausible growth and diffusion, they do not guarantee preservation of global anatomical topology. To prevent unrealistic predictions, such as fragmented white matter tracts or disconnected cortical gray matter, we introduce a topology-aware regularization based on centerline dice loss Shi et al. (2024) and no-overlap clause Nandanwar & Murty (2018).

**Definition 3** (Anatomy Structure Regularizer). *For selected classes $k \in \mathbb{A}_k$, we preserve topological consistency via the cl-Dice loss between the predicted soft mask $p_k(x, t)$ and ground-truth $q_k(x)$. Let $S(\cdot)$ denote a soft skeletonization operator. The predicted ($\mathbb{X}$) and ground-truth ($\mathbb{Y}$) soft skeletons are represented as:*

$$\mathbb{X}(p_k, q_k) = \frac{\langle S(p_k), \, q_k \rangle}{\langle S(p_k), \, \mathbf{1} \rangle + \varepsilon}, \qquad \mathbb{Y}(p_k, q_k) = \frac{\langle S(q_k), \, p_k \rangle}{\langle S(q_k), \, \mathbf{1} \rangle + \varepsilon},$$

*and the clDice loss*

$$\mathcal{L}_{\text{clDice}}(p_k, q_k) = 1 - \frac{2 \, \mathbb{X}(p_k, q_k) \, \mathbb{Y}(p_k, q_k)}{\mathbb{X}(p_k, q_k) + \mathbb{Y}(p_k, q_k) + \varepsilon}.$$

**Definition 4** (Anatomy Overlap Regularizer). *To encourage mutual exclusivity among classes, we penalize pairwise overlaps of the soft probabilities:*

$$\mathcal{L}_{\text{overlap}}(p) = \frac{2}{\mathbb{A}(\mathbb{A} - 1)} \sum_{1 \leq i < j \leq K} \mathbb{E}_x \big[ p_i(x, t) \, p_j(x, t) \big].$$

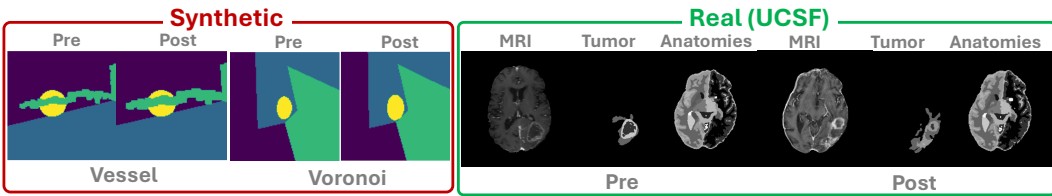

Figure 3: **Datasets.** We use two synthetic datasets and a real clinical dataset.

where $\mathbb{A}$ is the total number of anatomical classes.

**Anatomy Topology Loss.** Combining Definitions 4 and 5, we present a combined anatomy topology loss (ATL) function. We use a weighted sum of structure and exclusivity regularizers:

$$\mathcal{L}_{\text{ATL}}(p) = \sum_{k \in \mathcal{K}_{\text{cl}}} \lambda_1 \, \mathcal{L}_{\text{clDice}}(p_k, q_k) \; + \; \lambda_2 \, \mathcal{L}_{\text{overlap}}(p).$$

## 2.6 TRAINING

For training, we use a combination of Dice loss and Anatomy Topology Loss (ATL). The Dice loss ensures accurate overlap between predicted and ground-truth anatomical and tumor regions, while ATL enforces topological correctness by preserving the structural integrity of anatomical compartments. The final loss is represented as:

$$\min_{\theta} \; \mathcal{L}(\theta) = \underbrace{\sum_{k \in \mathbb{A}} \mathcal{L}_{\text{seg}}(p^{\text{pred}}, p^{\text{gt}})}_{\text{Segmentation loss}} + \lambda_{\text{topo}} \underbrace{\sum_{k \in \mathbb{A}} \mathcal{L}_{\text{ATL}}(p_k)}_{\text{Topology loss}} + \lambda_{\text{reg}} \|\theta\|^2 \qquad (3)$$

.

## 3 EXPERIMENTS AND RESULTS

### 3.1 EXPERIMENTAL SETUP

**Datasets.** We first validate our approach on two synthetic benchmarks designed for stress-testing: a Voronoi-based dataset and a vessel-tree dataset (Figure 3). These controlled settings allow us to systematically evaluate stability and robustness. Additional implementation and generation details are provided in the Appendix A. We further evaluate our framework on a clinical dataset, UCSF-ALPTDG dataset Fields et al. (2024), which provides multi-timepoint imaging of adult brain gliomas.

**Baselines.** We compare our method against two broad categories of approaches. The first includes deep learning–based models such as U-Net (Ronneberger et al., 2015), ConvLSTM (Shi et al., 2015), and NeuralODE Chen et al. (2018), which represent widely used architectures for image segmentation and temporal modeling. The second comprises PDE-driven formulations, namely Fisher-KPP (Fisher, 1937; Kolmogorov, 1937), and Cross-diffusion (Vanag & Epstein, 2009) models, which provide interpretable mechanistic baselines rooted in tumor growth dynamics. All these models were tuned to take pre-treatment multi-anatomy and tumor labels as input along with the treatment variables and to predict the subsequent changes in both anatomical structures and tumor regions in the post-treatment images. This comparison allows us to evaluate performance relative to both data-driven black-box methods and theory-guided physics-based models.

**Evaluation metrics.** For evaluation of the generated tumor masks, we use Dice-Sørensen Coefficient (DSC) and Hausdorff Distance-95th percentile (HD95). For all experimentations, we use a 5-fold cross-validation strategy and report the average performance on 5 folds.

### 3.2 CLINICAL APPLICATIONS.

**Quantitative analysis.** In Table 1, we report DSC and HD95 scores for anatomy and tumor mask segmentation tasks. U-Net and ConvLSTM reached DSCs of $63.57 \pm 2.29$ and $64.60 \pm 2.12$

Table 1: **Post-treatment anatomy and tumor mask prediction**. We compare our method with different DL and PDE-based baselines. DSC and HD95 Metrics reported as $\mu \pm \sigma$. Best results in **bold** and second best in *underline*.

| Method | Voronoi DSC | Voronoi HD95 | Vessel DSC | Vessel HD95 | UCSF DSC | UCSF HD95 |
|---|---|---|---|---|---|---|
| UNet | $66.08 \pm 9.82$ | $29.48 \pm 7.67$ | $60.31 \pm 2.16$ | $36.70 \pm 2.84$ | $63.57 \pm 2.29$ | $17.87 \pm 6.10$ |
| ConvLSTM | $69.86 \pm 3.96$ | $26.16 \pm 2.38$ | $59.62 \pm 4.09$ | $37.11 \pm 3.22$ | $64.60 \pm 2.12$ | $14.29 \pm 4.32$ |
| NeuralODE | $69.01 \pm 11.43$ | $28.18 \pm 6.73$ | $62.90 \pm 3.40$ | $33.98 \pm 4.12$ | $58.00 \pm 9.20$ | $27.12 \pm 19.88$ |
| Fisher-KPP | $71.57 \pm 0.34$ | $38.11 \pm 0.30$ | $69.04 \pm 0.68$ | $27.50 \pm 1.17$ | $59.88 \pm 1.36$ | $16.35 \pm 3.35$ |
| Cross-diffusion | $71.53 \pm 0.35$ | $38.11 \pm 0.29$ | $66.32 \pm 0.94$ | $37.63 \pm 1.65$ | $63.22 \pm 1.96$ | $10.49 \pm 1.98$ |
| Ours | $\mathbf{95.70 \pm 0.30}$ | $\mathbf{1.56 \pm 0.14}$ | $\mathbf{71.14 \pm 0.25}$ | $\mathbf{19.35 \pm 1.12}$ | $\mathbf{65.37 \pm 0.35}$ | $\mathbf{10.22 \pm 0.67}$ |

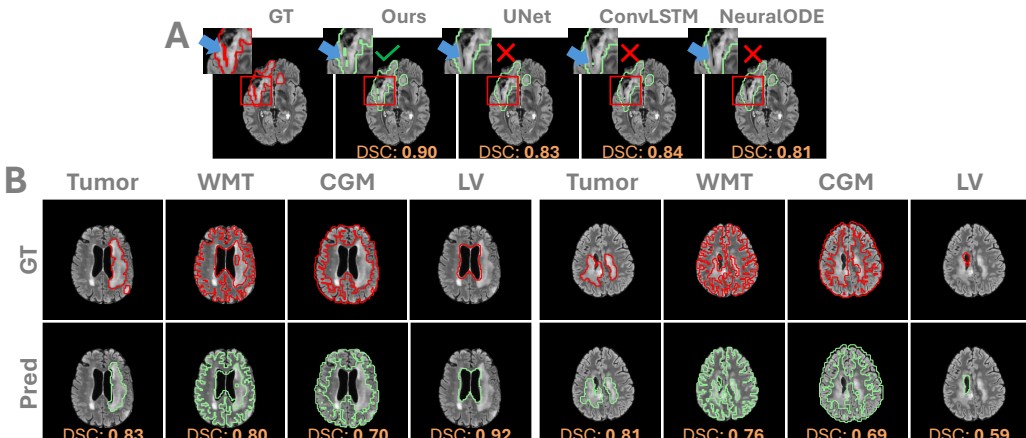

Figure 4: **Qualitative results.** A. We compare the tumor segmentation masks from our generated method with different baselines. We report the DSC scores for each cases. B. We show the ground-truth (in red) and predicted (in green) tumor, white matter tracts, cortical gray matter, and lateral ventricle segmentation contours. We report the DSC score for each structure.

with HD95 of $17.87 \pm 6.10$ and $14.29 \pm 4.32$, respectively, while NeuralODE performed worse ($58.00 \pm 9.20, 27.12 \pm 19.88$). Fisher–KPP achieved a DSC $\approx 59.88 \pm 21.36$ with HD95 $16.35 \pm 3.35$, and cross-diffusion acheived ($63.22 \pm 1.96, 10.49 \pm 1.98$). Our method surpassed all baselines with the highest DSC ($65.37 \pm 0.35$) and lowest HD95 ($10.22 \pm 0.67$), demonstrating both accuracy and stability. Clinically, improved boundary precision reduces uncertainty in treatment planning, while the lower variance indicates robustness that is crucial for consistent deployment across heterogenous patient populations. Additionally, in Figure 5 A, we compare the DSC of the proposed method with different baselines for different anatomies. In Figure 5 B, we report cl-Dice for computing the topological accuracy of the generated anatomical structures. We observe that Anatomy-DT achieved higher DSC and cl-Dice for both anatomies compared to the baselines. In summary, we conclude that our proposed method achieves the best performance on both combined segmentation and individual anatomy segmentation tasks.

**Qualitative analysis.** In Figure 4 B, we show the ground truth and predicted tumor and anatomy masks for four different patients overlaid on top of the FLAIR sequence. We observe that our proposed method consistently achieved high DSC across different tumor and anatomies. In Figure 4 A, we compare the generated tumor and different anatomies from our method with different baselines like UNet, ConvLSTM, and NeuralODE. We provide zoomed-in views of the critial anatomy regions. We observe that our method accurately captures more granular patterns compared to the baselines.

### 3.3 TOY DATASETS

**Quantitative analysis.** Table 1 reports segmentation performance on the toy Voronoi and Vessel datasets. On the Voronoi dataset, our method achieves a DSC of $0.9570 \pm 0.0030$ and HD95 of $1.56 \pm 0.14$, a ∼34% gain in overlap and ∼94% reduction in boundary error compared to the

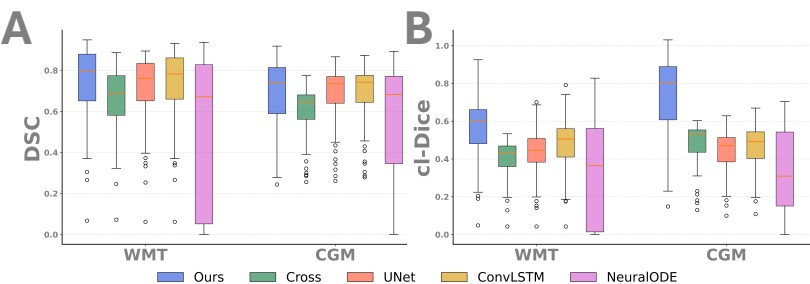

Figure 5: **Results on different structures.** We show box plots for DSC and cl-Dice scores of different structures (WMT and CGM).

Table 2: Ablation study on spatial regularizer, growth CNN, and topology loss.

| Spatial | Growth CNN | Topology | DSC $(\mu \pm \sigma)$ | HD95 $(\mu \pm \sigma)$ |
|---|---|---|---|---|
| ✗ | ✓ | ✗ | $64.82 \pm 4.99$ | $40.27 \pm 2.22$ |
| ✓ | ✗ | ✗ | $70.53 \pm 0.45$ | $20.23 \pm 0.29$ |
| ✓ | ✗ | ✓ | $70.50 \pm 0.44$ | $20.23 \pm 0.30$ |
| ✓ | ✓ | ✓ | $\mathbf{95.70 \pm 0.30}$ | $\mathbf{1.56 \pm 0.14}$ |

best baseline. PDE-only models (Fisher, Cross-diffusion) show consistent DSC of around 0.715 but very poor HD95 ($\sim$38), indicating coarse front propagation without fine boundary fidelity. On Vessel dataset, which emphasizes thin and filamentary structures, our method attains DSC of $0.7114 \pm 0.0025$ and HD95 of $19.35 \pm 1.12$. This marks a $\sim$7.3% DSC gain over Cross-diffusion and a $\sim$43% HD95 reduction versus NeuralODE. The gains are most evident in HD95, reflecting improved adherence to elongated vascular boundaries.

Overall, the toy results show that PDE-only models capture broad dynamics but miss fine structures, while learning-only baselines underperform on topology-sensitive geometries. Our method consistently improves both overlap and boundary metrics with low variance across folds.

**Stability and Sensitivity Analyses.** We conducted a series of controlled ablations to evaluate the stability and robustness of our cross-diffusion PDE model. Varying the integration step size $\Delta t$ while fixing the rollout horizon $T \approx 1.0$ revealed a clear trade-off: smaller steps achieved high Dice (0.9584 at $\Delta t = 0.1$) and low HD95, whereas large steps led to numerical collapse (Dice 0.7851 at $\Delta t = 0.3$). Analysis of the Jacobi solver confirmed diminishing returns beyond 4 iterations, with optimal accuracy obtained at 1–2 iterations and only marginal degradation at higher counts; sweeping the relaxation factor $\omega$ around the default ($\omega = 0.8$–1.0) showed minimal sensitivity. Resolution scaling highlighted strong performance up to $128 \times 128$, but severe degradation at larger grids, consistent with increased stiffness of the PDE operator. Sweeps over cross-diffusion strength $\chi$, tumor carrying capacity $\hat{C}$, and TV regularization $\lambda_{tv}$ all produced stable, near-constant Dice around 0.953, with slight improvements at moderate regularization. Finally, constraining the spatial growth rate ($k_{\max}$) demonstrated a sharp stability boundary: performance peaked at $k_{\max} = 2.0$ (Dice 0.9540), but collapsed when overly permissive ($k_{\max} = 10.0$, Dice 0.61). Collectively, these results delineate the regime where the proposed PDE model yields robust, high-fidelity forecasts while exposing failure modes outside stable parameter ranges.

**Ablation Analysis.** To disentangle the contributions of spatial modeling and topology constraints, we conducted a controlled ablation study (Table 2). Removing spatial features led to a substantial performance drop: DSC decreased to $64.82 \pm 4.99$ and HD95 increased to $40.27 \pm 2.22$. Introducing spatial dynamics (without Growth CNN) yielded the largest gain, improving Dice to $70.53 \pm 0.45$ and reducing HD95 by nearly half ($20.23 \pm 0.29$), showcasing the importance of spatially aware growth modeling. Adding the topology loss maintained competitive Dice ($70.50 \pm 7.64$) while preserving boundary coherence, as reflected in a moderate HD95 ($20.23 \pm 0.30$). We observe that combining Growth CNN along with the spatial terms and topology loss achieved the best performance (DSC: $95.7 \pm 0.3$ and HD95: $1.56 \pm 0.14$).

## 3.4 DISCUSSIONS

Our findings show that integrating physics-informed modeling with data-driven learning consistently improves both segmentation accuracy and stability. On synthetic benchmarks, our method outperformed PDE-only baselines, which captured coarse dynamics but failed at fine boundaries, and deep learning models, which struggled on topology-sensitive structures. Sensitivity analyses further confirmed that our PDE backbone remains robust across solver configurations, with clear stability limits at extreme parameter ranges. On the clinical UCSF dataset, our approach achieved the highest DSC and lowest HD95 with reduced variance, corroborating its robustness and clinical applicability. Clinically, improved boundary fidelity can directly reduce uncertainty in radiotherapy margin definition and surgical planning, thereby enhancing treatment precision. The findings demonstrate that our framework exhibits strong robustness and holds significant promise for translation into clinical practice.

## 4 RELATED WORK

**Generative Modeling in Medical Imaging.** Deep generative models such as GANs, VAEs, and diffusion processes have become standard tools for medical image synthesis, translation, and augmentation (Isola et al., 2017; Kingma et al., 2016; Ho et al., 2020; Dhariwal & Nichol, 2021). Several works extend these methods to clinical contexts, including modality transfer and reconstruction (Chartsias et al., 2017; Armanious et al., 2020), and more recently, diffusion-based synthesis (Özbey et al., 2023; Kim & Park, 2024). Longitudinal prediction—generating a follow-up exam from a baseline—has been pursued with conditional generative models or deformation fields derived from registration networks (Jie et al., 2016; Qin et al., 2019). However, these approaches typically emphasize pixel fidelity rather than anatomical validity; they may produce overlapping tissues, disconnected lobes, or topologically implausible vessels. A few recent works use topological constraints for guiding diffusion models (Bhattacharya et al., 2025a; Gupta et al., 2024; Xu et al., 2025). Our work differs by embedding biological priors directly in the generative process: tissues are constrained to evolve within a cross-diffusion PDE on the probability simplex, with topology regularized explicitly via persistent homology.

**Digital Twins in Oncology.** DT technology in oncology has gained prominence for its potential to model tumor growth Enderling & AJ Chaplain (2014); Oden et al. (2010) and predict treatment responses Lal et al. (2020); Chaudhuri et al. (2023); Sun et al. (2023). Early models utilized cellular automata Mallet & De Pillis (2006); Moreira & Deutsch (2002) and reaction-diffusion equations Weis et al. (2015); Gatenby & Gawlinski (1996); Konukoglu et al. (2009) to simulate tumor proliferation and response to therapies. Recent advancements have integrated these mechanistic models with machine learning techniques, enabling more accurate predictions by incorporating longitudinal imaging and multi-omics data. For instance, the TumorTwin Kapteyn et al. (2025) framework offers a modular approach to constructing patient-specific DTs, facilitating the simulation of tumor dynamics and treatment outcomes. Additionally, predictive DTs have been developed to optimize radiotherapy planning by accounting for spatially varying tumor characteristics and treatment uncertainties Chaudhuri et al. (2023). Despite these advancements, challenges remain in low data resources and complex treatment modeling pipelines, which are critical for the clinical application of DTs in oncology.

## 5 CONCLUSION

We proposed a principled framework for anatomy-to-anatomy evolution that unites cross-diffusion PDE dynamics, differentiable IMEX solvers, and topology-preserving regularization within a deep learning setting. By formulating tissue evolution as a probability field constrained to the simplex, our approach enforces exclusivity and inter-class competition, while persistent homology ensures global structural validity. Theoretical guarantees of weak-solution existence, unconditional stability, and simplex invariance distinguish our method from conventional generative models. Anatomy-DT's PDE–deep learning formulation offers a mathematically grounded path toward clinically meaningful digital twins that simulate disease trajectories in a stable, interpretable, and topologically consistent manner.

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
