# APPENDIX

Here, we provide additional details on datasets, additional results and theoretical proofs.

## A  DATASETS

For quantitative comparisons and stress-testing, we synthesize two datasets namely Voronoi and Vessel. Details of these datasets are provided here:

**Voronoi.** The Voronoi dataset generates multi-class smooth, organ-like regions using soft Voronoi partitions, where each class is formed by placing random sites and assigning pixels based on distance (shown in Figure 6 A). This creates natural-looking multi-organ structures with smooth boundaries. An additional tumor class is seeded near the center and modeled as a Gaussian blob, which expands at the second timepoint while slightly displacing its neighboring tissue. The dataset is well suited for testing models on smooth anatomical boundaries and tumor–organ interactions.

**Vessel.** The Vessel dataset focuses on vascular structures. It first builds a branching random-walk skeleton to mimic a vessel tree, then dilates it to obtain thickened vessels (shown in Figure 6 B). Surrounding regions are filled with soft Voronoi partitions to represent lobes, and a tumor is seeded near a vessel, reflecting biological tendencies of tumor growth along vasculature. Between the two timepoints, vessels thicken and the tumor grows preferentially along vessel proximity, creating realistic dynamics. Together, Voronoi and VesselTree datasets provide complementary challenges: one emphasizes smooth organ partitions, while the other emphasizes branching topology and vessel-guided tumor growth.

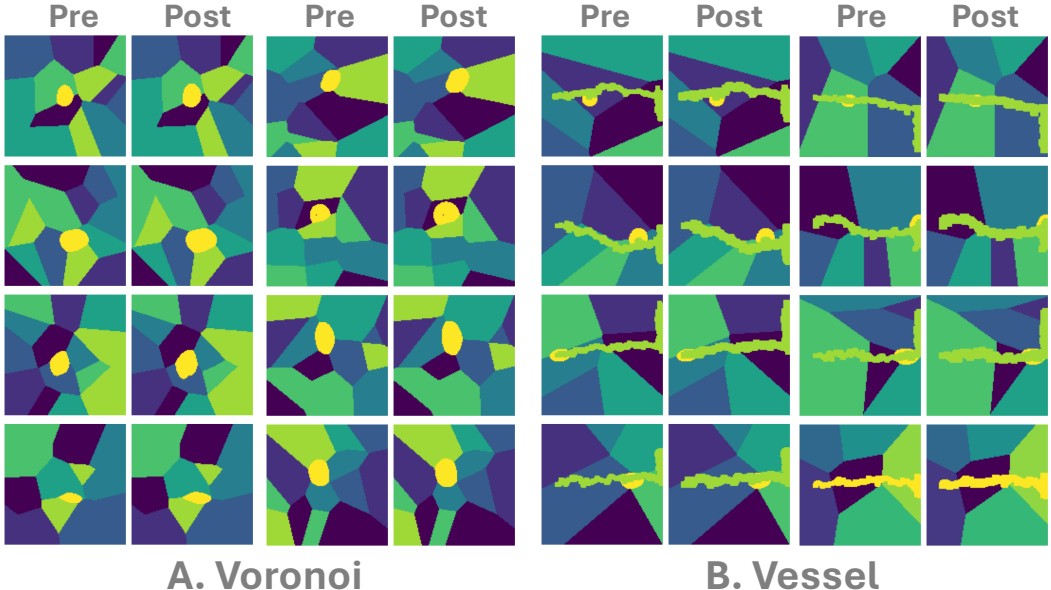

Figure 6: Additional examples of the toy datasets: Voronoi (A) and Vessel (B).

## B  ADDITIONAL RESULTS

**Sensitivity analysis.** The DSC and HD95 scores are reported for different Anatomy-DT parameters in Figure 7.

**Qualitative results.** Additional results are shown Figure 8.

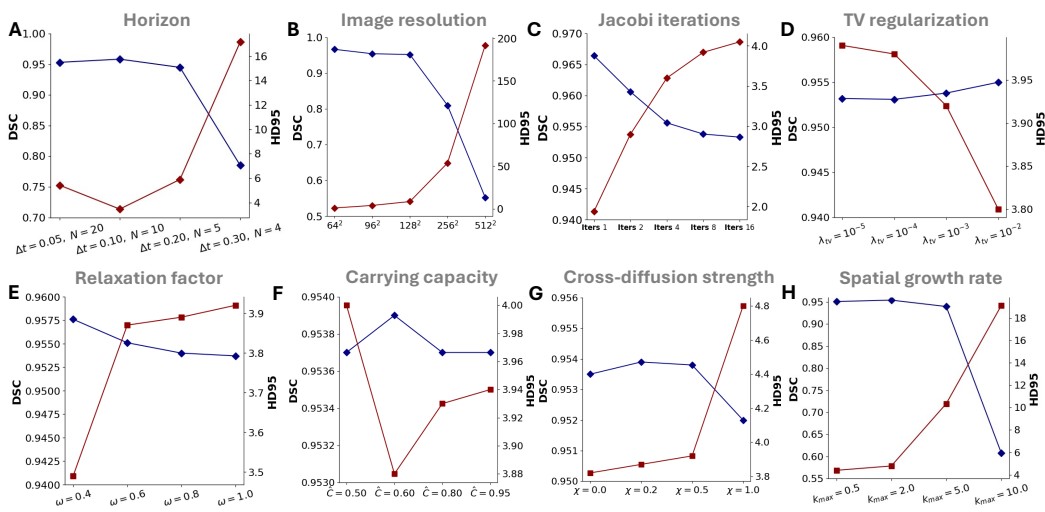

Figure 7: Sensitivity analysis.

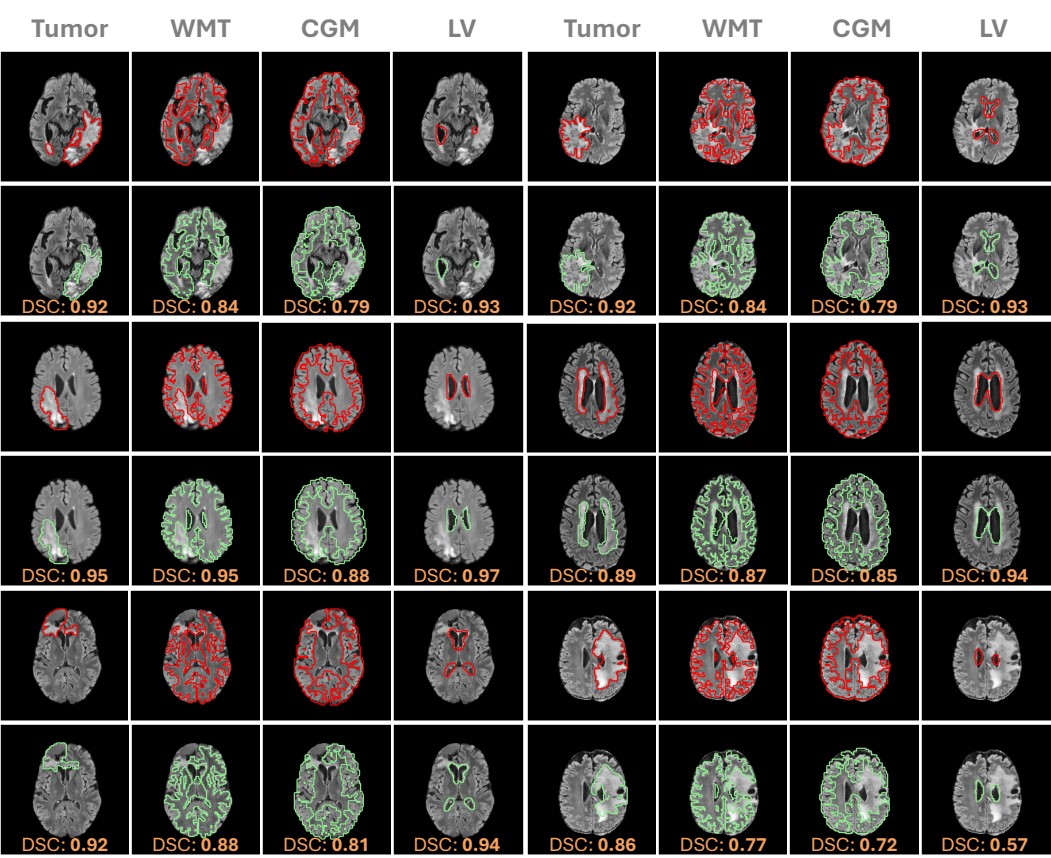

Figure 8: Additional qualitative results.

## C    THEORETICAL GUARANTEES

**Theorem 1** (Existence and Uniqueness). *Under standard assumptions on diffusion ($D_k$ coercive), cross-diffusion ($\chi_{kj}$ bounded), and Lipschitz reaction terms $r_k$, the PDE system in equation 2 admits a unique weak solution $p(x, t)$ on $[0, T]$.*

**Theorem 2** (Stability of IMEX Scheme). *The implicit treatment of diffusion and cross-diffusion ensures unconditional stability with respect to step size $\Delta t$. The explicit treatment of reaction and intervention terms preserves discontinuities introduced by clinical events.*

**Lemma 1** (Feasibility and Conservation). *Projection to the simplex $\Delta^{K-1}$ guarantees $p_k(x, t) \geq 0$ and $\sum_{k=1}^{K} p_k(x, t) = 1$ for all $t$. Thus, the solution remains a valid probabilistic anatomical state throughout integration.*

**Theorem 3** (Topology Preservation via ATL). *Let $\mathcal{L}_{\text{ATL}}(p) = \lambda_1 \sum_{k \in \mathcal{K}_{\text{cl}}} \mathcal{L}_{\text{clDice}}(p_k, q_k) + \lambda_2 \mathcal{L}_{\text{overlap}}(p)$ with $\lambda_1, \lambda_2 > 0$. Assume $S(\cdot)$ is the soft skeletonization used in clDice and satisfies the standard properties in* Shi et al. (2024): *continuity, idempotence on 1-pixel wide skeleta, and morphological thinning consistency. Then any minimizer $p^\star$ of $\mathcal{L}_{\text{ATL}}$ over feasible probability fields satisfies: (i) for each $k \in \mathcal{K}_{\text{cl}}$, the predicted and reference skeleta coincide (connectivity/homotopy preserved), and (ii) $p_i^\star(x) p_j^\star(x) = 0$ a.e. for all $i \neq j$ (mutual exclusivity).*

*Proof.* **(A) clDice term enforces skeleton (connectivity) agreement.** For a fixed class $k$, clDice is

$$\mathcal{L}_{\text{clDice}}(p_k, q_k) = 1 - \frac{2 \langle S(p_k), q_k \rangle}{\langle S(p_k), \mathbf{1} \rangle + \langle S(q_k), \mathbf{1} \rangle + \varepsilon},$$

which equals 0 iff the two directional skeleton precisions/recalls are 1, i.e., $S(p_k) \subseteq q_k$ and $S(q_k) \subseteq p_k$ up to null sets. Under the assumptions on $S(\cdot)$, this yields equality of skeleta and, by the clDice analysis, preservation of connectivity up to homotopy for binary segmentations in 2D/3D. Thus, for any $\lambda_1 > 0$, the ATL minimizer must satisfy $S(p_k) = S(q_k)$ for all $k \in \mathcal{K}_{\text{cl}}$. $\square$

**(B) Overlap term enforces exclusivity.** The overlap penalty

$$\mathcal{L}_{\text{overlap}}(p) = \frac{2}{K(K-1)} \sum_{1 \leq i < j \leq K} \mathbb{E}_x \big[ p_i(x) p_j(x) \big]$$

is nonnegative and equals 0 iff $p_i(x) p_j(x) = 0$ a.e. for all $i \neq j$. Hence at any minimizer with $\lambda_2 > 0$ we have pairwise exclusivity everywhere. This is consistent with standard exclusion/non-overlap losses used to enforce disjoint organs in multi-class medical segmentation.

**(C) Combined ATL.** Since both terms are nonnegative, the minimum of $\mathcal{L}_{\text{ATL}}$ is achieved only when each term attains its minimum. Therefore a minimizer $p^\star$ simultaneously satisfies (A) and (B): skeleton (connectivity) agreement for selected classes and mutual exclusivity across all classes. $\square$