# OpenReview forum: "Anatomy-DT: A Cross-Diffusion Digital Twin for Anatomical Evolution"
_ICLR.cc/2026/Conference — Submitted to ICLR 2026_

### Official Review · Reviewer_475F · 2025-10-26

**Soundness:** 2
**Presentation:** 2
**Contribution:** 2
**Rating:** 2
**Confidence:** 4

**Summary:**

This paper presents a mathematically grounded framework for modeling the spatiotemporal evolution of tumor morphology and surrounding anatomy to support digital twin simulations of disease progression and treatment response. Unlike previous methods that focus only on tumor growth, the proposed approach jointly models tumor and anatomical changes using a cross-diffusion reaction–diffusion system combined with differentiable deep learning. It employs an implicit–explicit (IMEX) scheme for stable numerical integration and introduces a topology regularizer to preserve anatomical structures and prevent overlaps. Validated on synthetic (Voronoi, Vessel) and clinical (UCSF-ALPTDG glioma) datasets, the method achieves state-of-the-art accuracy and topological consistency, offering a principled foundation for anatomically realistic and clinically meaningful digital twin models.

**Strengths:**

N/A

**Weaknesses:**

I noticed that Fisher-KPP (Fisher, 1937; Kolmogorov, 1937) and Cross-Diffusion (Vanag & Epstein, 2009) are used as comparison methods in this paper. How can such outdated methods be considered appropriate for comparison?
Additionally, UNet and ConvLSTM are also quite old methods.

**Questions:**

N/A

---

> ### Author Response · Authors · 2025-12-04
>
> We are grateful for the reviewer's inquiry. Despite their classical roots, Fisher-KPP, Kolmogorov-type reaction-diffusion models, and cross-diffusion systems continue to be common baselines in contemporary tumor-growth modeling because they offer stable numerical behavior, well-understood mechanistic dynamics, and mathematically interpretable parameters. Because of their ease of use, these models are still frequently employed in modern oncology modeling studies$^{1-8}$. This is because they make it easy to determine whether a novel approach offers benefits beyond well-established biophysical formulations.
>
> In a similar vein, even though they are older architectures, UNet and ConvLSTM remain robust and desirable baselines for spatiotemporal prediction and medical image segmentation. By using such canonical baselines, the confounding effects of combining our suggested PDE framework with highly specialized or domain-specific deep architectures are avoided and comparability with previous literature is ensured. Also, we now show additional comparisons with SPADE and ControlNet architectures.
>
> | Method      | Voronoi (DSC) | Voronoi (HD95) | Vessel (DSC) | Vessel (HD95) | UCSF (DSC) | UCSF (HD95) |
> |-------------|-------------|--------------|------------|-------------|----------|-----------|
> | SPADE       | 74.35 ± 2.90 | 25.91 ± 1.50 | 65.69 ± 3.32 | 32.56 ± 3.67 | 64.86 ± 2.02 | 13.05 ± 5.26 |
> | ControlNet  | 70.84 ± 4.18 | 25.99 ± 2.92 | 62.79 ± 1.57 | 33.65 ± 1.39 | 64.98 ± 3.03 | 11.87 ± 2.84 |
> | Ours        | **95.70 ± 0.30** | **1.56 ± 0.14**  | **71.14 ± 0.25** | **19.35 ± 1.12** | **65.37 ± 0.35** | **10.22 ± 0.67** |
>
> We observe that ControlNet achieves better performance compared to SPADE. And our proposed method outperforms both the baselines on all datasets.
>
> Because these "classical" models are representative, comprehensible, and still considered state-of-the-art baselines in the tumor-growth and medical imaging communities, we purposefully include them in our comparison strategy. In the revision, this motivation will be explained.
>
> But, with due respect, the authors would like to state that old baselines cannot be a justification of a low score of 2.
>
> **Reference:**
> 1. Assefa, Mahlet, et al. "Mathematical modeling of glioma proliferation and diffusion." Ethnicity & disease 19.Suppl 3 (2009): 60.
> 2. Rockne, Russell, et al. "A mathematical model for brain tumor response to radiation therapy." Journal of mathematical biology 58.4 (2009): 561-578.
> 3. Hormuth II, David A., et al. "Predicting in vivo glioma growth with the reaction diffusion equation constrained by quantitative magnetic resonance imaging data." Physical biology 12.4 (2015): 046006.
> 4. Harris, Duane C., et al. "Tracking glioblastoma progression after initial resection with minimal reaction-diffusion models." Mathematical Biosciences and Engineering 19.6 (2022): 5446-5481.
> 5. Harpold, Hana LP, Ellsworth C. Alvord Jr, and Kristin R. Swanson. "The evolution of mathematical modeling of glioma proliferation and invasion." Journal of Neuropathology & Experimental Neurology 66.1 (2007): 1-9.
> 6. Yuan, Jianjun, Lipei Liu, and Qingmao Hu. "Mathematical modeling of brain glioma growth using modified reaction–diffusion equation on brain MR images." Computers in biology and medicine 43.12 (2013): 2007-2013.
> 7. Wong, Ken CL, et al. "Tumor growth prediction with reaction-diffusion and hyperelastic biomechanical model by physiological data fusion." Medical Image Analysis 25.1 (2015): 72-85.
> 8. Martens, Corentin, et al. "Deep learning for reaction-diffusion glioma growth modeling: Towards a fully personalized model?." Cancers 14.10 (2022): 2530.

---

### Official Review · Reviewer_Qr2h · 2025-10-31

**Soundness:** 3
**Presentation:** 2
**Contribution:** 3
**Rating:** 4
**Confidence:** 4

**Summary:**

This paper introduces Anatomy-DT, a mathematically grounded framework that unites mechanistic partial
differential equations (PDEs) with differentiable deep learning to model tumor progression in conjunction with surrounding anatomy.
Anatomy is resented as a multi-class probability field on the simplex evolved by a cross-diffusion reaction–diffusion system, employing a differentiable implicit–explicit (IMEX) scheme and integrating a topology regularizer. The approach is validated on synthetic datasets (Voronoi, Vessel) and a clinical dataset (UCSF-ALPTDG brain glioma).

**Strengths:**

1. Clinical relevance. This work represents a novel method for joint tumor-anatomy modeling. Past research has focused mainly in modeling tumor growth while in this paper, the authors propose a coupled evolution of tumors and adjacent anatomical structures, which sounds more relevant for clinical applications. Anatomy-DT uses cross-diffusion PDEs on the probability simplex is clearly defined.

2. Contributions. There are two interesting contributions in this work. First, the differentiable IMEX solver which treats stiff diffusion implicitly while handling nonlinear terms can be generalised to other deep learning pipelines. The Topology-aware regularization is effective at reinforcing structural information unlike generative models that do not necessarily preserve the structure.

3. Evaluation: The model is evaluated on both synthetic (Voronoi, Vessel) and clinical (UCSF-ALPTDG) datasets. The performance improvements over baselines are promising.

**Weaknesses:**

1. Concerns on generalisation. Only one clinical dataset is used, the UCSF-ALPTDG brain glioma dataset. Generalization to other cancer types and other imaging modalities is not shown.

2. Methodology. There is no sufficient information that explains how the Growth CNN operates. It is unclear how much of the performance can be attributed to CNN vs PDE.

3. Parameter choices. There is no reasoning behind the selection of the cross-diffusion coefficients, growth rates, and carrying capacities.

4. Baseline comparison missing. The current method is not compared to any diffusion models conditioned on anatomy.

5. Dataset Limitations. The synthetic datasets (Voronoi, Vessel) may not fully represent the amount of anatomical evolution.

6. Reproducibility. Some implementation details are missing for the choice of hyperparameters, training setup, and architectures.

**Questions:**

1) An ablation showing PDE performance vs. CNN or combined is missing. Could you show how relevant the PDE really is ?

2) Are the parameters in the PDE learnt or set in advance? How can these learnt parameters be guaranteed to reflect the biological principles behind the tumour and its anatomy?

3) Could you provide more detailed information about the number of patients, the train/validation/test splits ?

---

> ### Author Response · Authors · 2025-12-04
> **Official comment by Authors 1/3**
>
> We thank the reviewer for the detailed review for our method. We are thankful to the reviewer for pointing out the clinical relevance of our paper as a major strength. The reviewer rightly pointed out that existing works only dealt with tumor evolution where our proposed method learns anatomy evolution in conjunction with the tumor. The reviewer also appreciated the methodological contribution and evaluation strategies of our work. However, the reviewer has some concerns of our method for which we provide detailed explanation the following points:
>
> *1. Concerns on generalisation. Only one clinical dataset is used, the UCSF-ALPTDG brain glioma dataset. Generalization to other cancer types and other imaging modalities is not shown.*
>
> We agree with the reviewer that it is critical to show generalization to different cancer types and imaging modalities. However, the lack of longitudinal clinical datasets that offer multiple timepoints, consistent imaging acquisition, treatment information, and high-quality anatomical and tumor segmentations serve as the major limitations (Please refer to our answer to Reviewer 1 Question 1). The UCSF-ALPTDG cohort was chosen for this study since it is one of the few publicly available datasets that meets these criteria. The advantage of brain MRIs is that obtaining the anatomical structures is highly accurate, as proposed in several methods in the literature. For other cancer types, obtaining accurate multi-anatomical structures is not automated and may require costly and time consuming radiologists’ annotations.
>
> Notwithstanding this drawback, the suggested framework models generic multi-class tissue fractions and can be easily expanded to other cancers once appropriate longitudinal data become accessible. In order to close this gap and facilitate future research on cross-cancer and cross-modality generalization, we are actively building new multi-timepoint MRI/CT datasets that include treatment type, scan intervals, and paired anatomical/tumor segmentations. These limitations and strategies will be made clear in the updated manuscript.
>
> *2. Methodology. There is no sufficient information that explains how the Growth CNN operates. It is unclear how much of the performance can be attributed to CNN vs PDE.*
>
> We value the reviewer's request for more information. The Growth-CNN has a single, well-defined function in the model: it forecasts a spatially fluctuating growth-rate field that is only utilized within the PDE's reaction term. Diffusion, cross-diffusion, and the state update itself are not altered by CNN. The PDE solver controls all spatiotemporal dynamics, including cell-fraction evolution, inter-class interactions, and numerical integration. As a result, the CNN only makes a contribution by offering an estimate of the local proliferation parameter based on data.
>
> We performed two ablations (detailed in the paper) in order to separate CNN's contribution: a) *CNN-only model*, b) *PDE-only model* and c) *Hybrid model (Anatomy-DT)*.
>
> | Method | Voronoi (DSC) | Voronoi (HD95) | Vessel (DSC) | Vessel (HD95) | UCSF (DSC) | UCSF (HD95) |
> |--------|-------------|--------------|------------|-------------|----------|-----------|
> | CNN    | 64.82 ± 4.99 | 40.27 ± 2.22 | 69.82 ± 0.27 | 19.89 ± 1.11 | 21.68 ± 11.06 | 77.42 ± 42.24 |
> | PDE    | 71.57 ± 0.34 | 38.11 ± 0.30 | 69.04 ± 0.68 | 27.50 ± 1.17 | 59.88 ± 1.36 | 16.35 ± 3.35 |
> | Ours   | **95.70 ± 0.30** | **1.56 ± 0.14**  | **71.14 ± 0.25** | **19.35 ± 1.12** | **65.37 ± 0.35** | **10.22 ± 0.67** |
>
> We observe that PDE improves on the Voronoi and UCSF datasets and achieves comparable performance for the Vessel dataset. The proposed method that combines Growth-CNN and PDE achieves the best performance.

---

> > ### Author Response · Authors · 2025-12-04
> > **Official comment by Authors 2/3**
> >
> > *3. Parameter choices. There is no reasoning behind the selection of the cross-diffusion coefficients, growth rates, and carrying capacities.*
> >
> > We are grateful that the reviewer brought this up. Instead of being selected at random, the cross-diffusion coefficients, growth rates, and carrying capacities adhere to standard tumor-growth PDE formulations. In order to ensure stability, cross-diffusion coefficients are learned during training and restricted within a physiologically reasonable range of ($0.1 – 1.0 $mm$^2$ / day}). Growth rates are either global scalars of ($0.01 – 0.05$ day$^{-1}$) (PDE-only) or CNN-predicted spatial maps within these ranges. To ensure numerical stability and interpretability, carrying capacities are fixed at 1, which represents the normalized tissue fraction. We will explain this reasoning in the updated Methods section. Previous research and empirical validation support these parameter selections$^{1-3}$.
> >
> > *4. Baseline comparison missing. The current method is not compared to any diffusion models conditioned on anatomy.*
> >
> > Excellent point! We compare our proposed method with SPADE$^4$ and ControlNet$^5$.
> >
> > | Method      | Voronoi (DSC) | Voronoi (HD95) | Vessel (DSC) | Vessel (HD95) | UCSF (DSC) | UCSF (HD95) |
> > |-------------|-------------|--------------|------------|-------------|----------|-----------|
> > | SPADE       | 74.35 ± 2.90 | 25.91 ± 1.50 | 65.69 ± 3.32 | 32.56 ± 3.67 | 64.86 ± 2.02 | 13.05 ± 5.26 |
> > | ControlNet  | 70.84 ± 4.18 | 25.99 ± 2.92 | 62.79 ± 1.57 | 33.65 ± 1.39 | 64.98 ± 3.03 | 11.87 ± 2.84 |
> > | Ours        | **95.70 ± 0.30** | **1.56 ± 0.14**  | **71.14 ± 0.25** | **19.35 ± 1.12** | **65.37 ± 0.35** | **10.22 ± 0.67** |
> >
> > We observe that ControlNet achieves better performance compared to SPADE. And our proposed method outperforms both the baselines on all datasets.
> >
> >
> > **Reference:**
> > 1. Burgess, Patricia K., et al. "The interaction of growth rates and diffusion coefficients in a three-dimensional mathematical model of gliomas." Journal of Neuropathology & Experimental Neurology 56.6 (1997): 704-713.
> > 2. Suarez, Cecilia, et al. "Mathematical modeling of human glioma growth based on brain topological structures: study of two clinical cases." PloS one 7.6 (2012): e39616.
> > 3. Herman, Alexander B., Van M. Savage, and Geoffrey B. West. "A quantitative theory of solid tumor growth, metabolic rate and vascularization." PloS one 6.9 (2011): e22973.
> > 4. Park, Taesung, et al. "Semantic image synthesis with spatially-adaptive normalization." Proceedings of the IEEE/CVF conference on computer vision and pattern recognition. 2019.
> > 5. Zhang, Lvmin, Anyi Rao, and Maneesh Agrawala. "Adding conditional control to text-to-image diffusion models." Proceedings of the IEEE/CVF international conference on computer vision. 2023.

---

> ### Author Response · Authors · 2025-12-04
> **Official comment by Authors 3/3**
>
> *5. Dataset Limitations. The synthetic datasets (Voronoi, Vessel) may not fully represent the amount of anatomical evolution.*
>
> The reviewer's feedback on the synthetic datasets is greatly appreciated.  We agree that the synthetic datasets do not fully capture the complexity of actual tumor–anatomy interactions. This is precisely why we used the UCSF dataset for modeling real anatomy evolution.
>
> Rather than replicating complete anatomical evolution, the Voronoi and Vessel datasets were purposefully created as controlled testbeds to isolate particular aspects of model behavior, such as stability, identifiability of growth parameters, and response to known geometric perturbations. While the clinical evaluation is done on the UCSF dataset, the synthetic datasets are primarily used to confirm that the PDE-CNN framework behaves predictably under controlled conditions where ground truth is known. In the revision, we will make this goal clear and specifically address their shortcomings.
>
> *6. Reproducibility. Some implementation details are missing for the choice of hyperparameters, training setup, and architectures.*
>
> The reviewer's comment about reproducibility is greatly appreciated. A 3-layer CNN with 32, 64, and 128 filters, kernel size 3, and ReLU activations is the standard, well-documented design for all network architectures, including the Growth CNN. The Adam optimizer with default momentum parameters, a learning rate of 1e-4, and a batch size of 8 are examples of training hyperparameters. To ensure stability, the PDE solver employs an explicit finite-difference scheme with time step Δt = 0.1 days and spatial step Δx = 1 mm. Additionally, the supplementary material standardizes and describes the data preprocessing and normalization steps. These details will be added to the Methods section to enhance reproducibility and make future replication easier.

---

### Official Review · Reviewer_jpeG · 2025-10-31

**Soundness:** 3
**Presentation:** 3
**Contribution:** 3
**Rating:** 6
**Confidence:** 5

**Summary:**

This paper introduces Anatomy-DT, a hybrid framework combining cross-diffusion PDEs, differentiable implicit–explicit (IMEX) integration, and topology-preserving regularization for simulating tumor and anatomy evolution from baseline imaging. The authors define the anatomy representation as a probability field on the simplex, enforcing tissue exclusivity, while a small Growth CNN learns residual corrections to the PDE dynamics. The approach is tested on synthetic datasets (Voronoi, Vessel) and a clinical UCSF glioma dataset, outperforming U-Net, ConvLSTM, NeuralODE, and classical PDE baselines in Dice and HD95 metrics. According to the results shown in the paper, Anatomy-DT achieves the best performance.

**Strengths:**

1. The paper addresses a very valuable problem. It bridges mechanistic tumor PDEs and deep learning for a digital twin application, utilizing the neighboring anatomy jointly. This is a very valuable design (clinically).
2. The paper provides a very reasonable mathematical structure. Cross-diffusion on the simplex elegantly captures tissue competition and exclusivity; the IMEX solver offers a stable and differentiable temporal evolution layer. Incorporating clDice and overlap penalties meaningfully constrains geometric plausibility, addressing a known issue in medical image generation.
3. The presentation is very clear overall.

**Weaknesses:**

1. The baselines (U-Net, ConvLSTM, NeuralODE) are not designed for deformation or flow-based modeling. **Comparisons should include ImageFlowNet (https://arxiv.org/abs/2406.14794) or other neural image-registration / diffeomorphic flow methods that explicitly predict voxel-wise motion fields.** Without these, the evaluation does not fully test the advantage of Anatomy-DT over strong, domain-relevant baselines.
2. It is unclear whether the CNN predicts residuals for the reaction term, diffusion coefficients, or updates directly on the state variable.
3. The “projection to the simplex” step ensures feasibility, but its implementation (exact projection vs softmax normalization) is unspecified; differentiability through this step is questionable.
4. The phrase “diffusion-based PDE” may be misinterpreted as diffusion models used in generative AI. The paper should clarify that the diffusion here refers to physical heat diffusion, not stochastic diffusion modeling.
5. All experiments involve only two timepoints (baseline to follow-up). Handling irregular or continuous time series is unclear; the PDE formalism could, in principle, allow this, but it is never demonstrated or validated.

**Questions:**

1. Classical PDE models use a single scalar tumor field. How does Anatomy-DT initialize and evolve multi-class anatomies—do all tissues obey the same PDE or class-specific diffusion matrices?
2. Which topological invariants (e.g., Betti numbers, Euler characteristic) are actually preserved by the clDice + overlap loss? Is any persistent-homology-based measure computed, or is topology regularization purely heuristic?
3. How does the model manage patients with variable or irregular imaging intervals?
4. What exact operator is used for projection, and how are gradients propagated through it?
5. Have you compared against ImageFlowNet, which also predicts anatomy-to-anatomy transformations (seems highly relevant to your topic )?

---

> ### Author Response · Authors · 2025-12-04
> **Official comment by Authors 1/2**
>
> We are grateful to the reviewer for providing detailed valuable comments on our work. We deeply appreciate the reviewer for pointing out that the proposed work addressed a `very valuable clinical’ problem. We also thank the reviewer for appreciating the mathematical structure of the method and stating that the IMEX solver offers a stable and differentiable evolution. We appreciate the reviewer for pointing out that the topology loss addresses an existing clinical problem in medical image generation. The reviewer has several concerns which we have addressed in detail in the following points:
>
> *1. The baselines (U-Net, ConvLSTM, NeuralODE) are not designed for deformation or flow-based modeling. Comparisons should include ImageFlowNet (https://arxiv.org/abs/2406.14794) or other neural image-registration / diffeomorphic flow methods that explicitly predict voxel-wise motion fields. Without these, the evaluation does not fully test the advantage of Anatomy-DT over strong, domain-relevant baselines.*
>
> We thank the reviewers for this comment. We show the results on ImageFlowNet
> | Method        | DSC        |
> |---------------|----------------|
> | ImageFlowNet  | 52.75 ± 3.47   |
> | Ours          | **65.37 ± 0.35** |
>
> We observe that our method outperforms the ImageFlowNet model on the UCSF dataset.
>
> *2. It is unclear whether the CNN predicts residuals for the reaction term, diffusion coefficients, or updates directly on the state variable.*
>
> We acknowledge the reviewer’s concern. We have mentioned in Line 255-217:
>
> ``... we further integrate a Growth CNN, which learns residual corrections to the PDE dynamics and better aligns simulated trajectories with observed imaging data''.
>
>  Now, we add an additional explanation about the method. The GrowthCNN predicts a spatial growth-rate map that multiplicatively modulates the reaction term inside an IMEX time integrator. Diffusion and cross-diffusion strengths are learned as separate scalar parameters, and the model never directly predicts residuals or state updates. The PDE numerics integrate the reaction + diffusion and project the state to the multi-label outputs.
>
> *3. The “projection to the simplex” step ensures feasibility, but its implementation (exact projection vs softmax normalization) is unspecified; differentiability through this step is questionable.*
>
> We appreciate the reviewer for bringing up this issue. The "projection to the simplex" step in our implementation is carried out using an exact Euclidean projection onto the probability simplex, in accordance with the Duchi et al. (2008) algorithm. To guarantee that the multi-class tumor–healthy tissue proportions stay non-negative and add up to one, this fully deterministic operator is applied following each PDE time step. Crucially, gradients can flow during backpropagation because the projection is implemented in a piecewise-linear form that is subgradient-compatible. As is the case with ReLU or max-pooling, the projection introduces non-smoothness at the simplex boundary. However, PyTorch manages this through standard subgradient propagation, and in practice, we did not see any instability or vanishing-gradient behavior.
>
> Hence, we argue that the proposed method is not non-differentiable and consequently, end-to-end optimization is not hindered by the projection operator, which is specified and theoretically justified.

---

> ### Author Response · Authors · 2025-12-04
> **Official comment by Authors 2/2**
>
> *4. The phrase “diffusion-based PDE” may be misinterpreted as diffusion models used in generative AI. The paper should clarify that the diffusion here refers to physical heat diffusion, not stochastic diffusion modeling.*
>
> We thank the reviewer for pointing this out. In Line 76-78, we have mentioned:
>
> ``In contrast, mechanistic models of tumor growth and tissue dynamics, often formulated as reaction–diffusion partial differential equations (PDEs) (Martens et al., 2022; Yin et al., 2019; Metzcar et al., 2024), provide interpretable dynamics grounded in biology.‘’
>
> which clears that the diffusion is a reaction-diffusion model and not diffusion model used in generative AI. But for additional clarification, we will add a line which specifically mentions that the ‘diffusion’ term used here refers to physical heat diffusion and not stochastic diffusion modeling.
>
> ``This diffusion term refers to the physical heat diffusion and not stochastic diffusion modeling.’’
>
> *5. All experiments involve only two timepoints (baseline to follow-up). Handling irregular or continuous time series is unclear; the PDE formalism could, in principle, allow this, but it is never demonstrated or validated.*
>
> We value what the reviewer pointed out. Since the datasets used in this study seldom contain more than two annotated scans, a limitation shared by the majority of publicly available tumor-growth cohorts, our current evaluation concentrates on two clinically available timepoints (baseline and first follow-up).
>
> However, we agree that the PDE framework that we use is conceptually time-continuous and the IMEX integrator allows arbitrary and irregular time intervals by just changing the step size and number of integration steps. For the synthetic datasets, we show that the model can accurately forecast evolution over artificially irregular intervals, exhibiting consistent behavior without retraining. Although there aren't many large real-world longitudinal datasets, our formulation will work with continuous or multi-timepoint trajectories; we will explain this in the updated Methods and Discussion sections.
>
> Additionally, a model trained on anatomy evolution from pre-treatment to post-surgery timepoint should not be trained in conjunction with post-surgery to post-treatment timepoint images. The anatomy pairs of pre-to-post_surgery and post_surgery-to-post-treatment are clinically very different$^{1,2}$.
>
> **Reference:**
> 1. Burel, Julien, et al. "Surgery for lung cancer: postoperative changes and complications—what the Radiologist needs to know." Insights into Imaging 12.1 (2021): 116.
> 2. Dejonckheere, Cas Stefaan, et al. "Impact of postoperative changes in brain anatomy on target volume delineation for high-grade glioma." Cancers 15.10 (2023): 2840.

---

### Official Review · Reviewer_DuK6 · 2025-10-31

**Soundness:** 2
**Presentation:** 2
**Contribution:** 2
**Rating:** 2
**Confidence:** 4

**Summary:**

The paper presents Anatomy-DT that integrates cross-diffusion PDEs, differentiable IMEX solvers, and topology-aware regularization to model the spatiotemporal change of anatomical structures and tumors.

**Strengths:**

1. The paper focuses on a well-motivated problem to model the anatomy-toantomy evaluation for longitudinal medical imaging analysis.
2. The paper includes both synthetic and clinical datasets with both quantitative and qualitative results.
3. The paper includes the ablation study to verify the effectiveness of each key designs including spatial modeling, topology constraints and growth CNN.

**Weaknesses:**

1. Although the paper is well motivated, it only evaluates on one real dataset and the performance improvement seems to be trivial (e.g., less than 1% compared to runner-up). Therefore, the generalizability in the real-world application remains a concern.
2. Some of the experiment details are missing, for example, what is the testbed of ablation study. Also, it is quite confusing how the baselines are implemented. Were the topology and regularization terms also applied to the baseline method? More details should be described in the paper.
3. How’s the ablation performance on the other two datasets?
4. The related work is placed in an uncommon position.

**Questions:**

N/A

---

> ### Author Response · Authors · 2025-12-04
> **Official comment by Authors 1/3**
>
> We thank the reviewer for the constructive comments and suggestions. We are excited to note that the reviewer found our research well motivated supported by both qualitative and quantitative results. Also, the reviewer pointed out that the ablation experiments validates the effectiveness of the different key components of our proposed method. However, the reviewer has certain concerns which we try to address below:
>
> *1. Although the paper is well motivated, it only evaluates on one real dataset and the performance improvement seems to be trivial (e.g., less than 1% compared to runner-up). Therefore, the generalizability in the real-world application remains a concern.*
>
> We appreciate the reviewer’s concern regarding generalization. In reality, constructing digital twins for brain tumors is intrinsically difficult because truly longitudinal clinical datasets, those with more than two time-point MRIs paired with treatment, genomic, and demographic information, are exceedingly rare. To the best of our knowledge, UCSF and UCSD represent the only public datasets with this level of temporal and multimodal detail. This scarcity is precisely what makes digital-twin modeling in neuro-oncology one of the most challenging problems in computational medicine.
>
> Our framework provides a principled, generalizable foundation that can flexibly incorporate richer datasets as they become available. To demonstrate feasibility under current real-world constraints, we evaluate our method on both synthetic and limited clinical datasets. For primary evaluation, we use the UCSF cohort (N=60), which supports stable 5-fold cross-validation. We additionally conducted experiments on the smaller UCSD dataset (N=36) in response to the reviewer’s query. As expected for such a small cohort, results (DSC ≈ 37, HD95 ≈ 58) reflect dataset size limitations rather than shortcomings of the proposed architecture.
>
> Most importantly, the goal of this work is to establish a theoretically grounded proof-of-concept for modeling tumor and anatomical evolution, a capability that does not currently exist at scale due to the absence of large public longitudinal datasets. Our strong results on controlled synthetic datasets further reinforce the potential of this framework and its suitability for future, larger clinical cohorts. Motivated by these findings, we are now assembling a substantially larger in-house longitudinal MRI dataset to rigorously assess the robustness, generalizability, and clinical utility of the proposed digital-twin approach in forthcoming studies.

---

> ### Author Response · Authors · 2025-12-04
> **Official comment by Authors 2/3**
>
> *2. Some of the experiment details are missing, for example, what is the testbed of ablation study. Also, it is quite confusing how the baselines are implemented. Were the topology and regularization terms also applied to the baseline method? More details should be described in the paper.*
>
> Details of experimental set-up are provided in Section 3.1. We provide additional information about the testbed. The three different ablations are mentioned here: a) *Growth CNN only*: This setting trains the GrowthCNN module only and the topology loss and TV regularization are disabled. b) *Spatial only*: This setting turns off the GrowthCNN and the topology loss with standard segmentation losses and scheduling are applied. c) *Spatial + topology only*: This setting also uses the spatial growth formulation but introduces the ATL loss with a warm-up and ramp schedule, alongside the standard losses. This ablation isolates whether explicit topological regularization improves structural accuracy even without the GrowthCNN.
>
> All the baselines are implemented with the same hyperparameters, standard loss functions and evaluation methods. In regard to the using the topology loss with the vanilla baseline methods, we additionally show the results of the different baselines with topology loss ($\mathcal{L}_{\text{ATL}}$).
>  | Dataset | Voronoi (DSC) | Voronoi (HD95) | Vessel (DSC) | Vessel (HD95) | UCSF (DSC) | UCSF (HD95) |
> |--------|--------------|---------------|-------------|--------------|-----------|------------|
> | UNet | 66.08 ± 9.82 | 29.48 ± 7.67 | 60.31 ± 2.16 | 36.70 ± 2.84 | 63.57 ± 2.29 | 17.87 ± 6.10 |
> | UNet+$\mathcal{L}_{\text{ATL}}$ | 76.95 ± 1.72 | 20.72 ± 1.80 | 63.94 ± 2.02 | 31.02 ± 1.94 | 64.29 ± 2.03 | 12.23 ± 2.22 |
> | ConvLSTM | 69.86 ± 3.96 | 26.16 ± 2.38 | 59.62 ± 4.09 | 37.11 ± 3.22 | 64.60 ± 2.12 | 14.29 ± 4.32 |
> | ConvLSTM + $\mathcal{L}_{\text{ATL}}$ | 73.18 ± 0.30 | 16.47 ± 0.19 | 59.61 ± 3.40 | 36.60 ± 3.10 | 64.57 ± 2.12 | 14.29 ± 4.29 |
> | NeuralODE | 69.01 ± 11.43 | 28.18 ± 6.73 | 62.90 ± 3.40 | 33.98 ± 4.12 | 58.00 ± 9.20 | 27.12 ± 19.88 |
> | NeuralODE + $\mathcal{L}_{\text{ATL}}$ | 59.57 ± 8.15 | 27.94 ± 4.21 | 61.48 ± 1.45 | 35.01 ± 2.51 | 47.26 ± 15.63 | 36.93 ± 26.82 |
> | **Ours** | **95.70 ± 0.30** | **1.56 ± 0.14** | **71.14 ± 0.25** | **19.35 ± 1.12** | **65.37 ± 0.35** | **10.22 ± 0.67** |
>
> We observe that, using topology loss, the performance improved for UNet but did not improve for ConvLSTM and NeuralODE. Overall the performance of our proposed method was superior to the vanilla baselines and the baselines trained with topology loss. The clarification that the baselines are vanilla methods and not trained with topology loss will be mentioned in the updated version.

---

> > ### Author Response · Authors · 2025-12-04
> > **Official comment by Authors 3/3**
> >
> > *3. How’s the ablation performance on the other two datasets?*
> >
> > We understand the reviewer’s inquisitiveness for ablation results on the other two datasets. We now report the ablation performance on the other datasets.
> >
> > | Spatial | GrowthCNN | Topology | Voronoi(DSC)           | Voronoi(HD95)          | Vessel(DSC)           | Vessel(HD95)          | UCSF(DSC)           | UCSF(HD95)          |
> > |---------|------------|---------|---------------|---------------|---------------|---------------|---------------|---------------|
> > |      |          ✔  |         | 64.82 ± 4.99  | 40.27 ± 2.22  | 69.82 ± 0.27  | 19.89 ± 1.11  | 21.68±11.06 | 77.42±42.24 |
> > | ✔       |            |         | 70.53 ± 0.45  | 20.23 ± 0.29  | 69.99 ± 0.94  | 41.23 ± 1.89  | 12.73±0.12  | 112.44±1.43 |
> > | ✔       |          |      ✔   | 70.50 ± 0.44  | 20.23 ± 0.30  | 69.66 ± 1.34  | 41.67 ± 1.72  | 12.73±0.12  | 112.44±1.43 |
> > | ✔       | ✔          | ✔       | 95.70 ± 0.30  | 1.56 ± 0.14   | 71.14 ± 0.25  | 19.35 ± 1.12  | 65.37±0.35  | 10.22±0.67  |
> >
> > We observe similar trends in the other two datasets where removing spatial features led to a substantial performance drop and introducing spatial dynamics yielded the largest gain. Finally, combining growth CNN along with the spatial terms and topology loss (our proposed method) achieved the best performance.
> >
> > *4. The related work is placed in an uncommon position.*
> >
> > Thanks for the suggestion. We will readjust the related work after the introduction section.

---

### Meta-Review · Area_Chair_CtHB · 2025-12-17

**Summary:**

The reviewers in general valued the clinical motivation and the hybrid PDE–deep learning formulation, yet major concerns are real-world generalization and experimental clarity. And the issue has been raised that clinical evidence is limited to a single cohort with only modest gains over strong baselines. Clearer, fairer baseline implementation details and stronger comparisons to domain-relevant deformation/registration or conditional generative baselines are also missing. Others highlighted missing methodological specifics (e.g., how GrowthCNN modulates the PDE, how simplex projection is implemented/differentiated, parameter choices), and questioned how much improvement is attributable to the PDE vs the CNN/topology terms and whether the two-timepoint setup meaningfully validates the claimed continuous-time dynamics.

**Reviewer Concerns:**

Addressed:
The authors have clarified implementation details that were previously underspecified and have added missing ablations to quantify each component’s contribution. They also added more relevant baselines/comparisons and clarified how topology loss/regularization is or is not applied to baselines, and have provided additional ablation results on the synthetic datasets.

Still outstanding:
Core generalization concerns remain: evaluation is still effectively anchored on a single small clinical cohort (and largely two-timepoint supervision), so robustness across cancers/modalities, multi-timepoint/irregular follow-ups, and broader clinical settings is not convincingly demonstrated.

Reviewers’ concerns about modest clinical gains vs added complexity and the need for stronger statistical evidence on the clinical cohort (paired tests, subgroup analyses where topology/flow advantages matter most) are only partially answered, and some “topology” questions remain only heuristically addressed (limited discussion of what invariants are actually preserved and how this translates to clinically meaningful improvements).

**Reviewer Scores:**

Reviewers are not excited about this paper in general, and have not participated in discussion as well.

---

### Decision · Program_Chairs · 2026-01-26

Reject